

# Effects of glucose, ethanol and acetic acid on regulation of ADH2 gene from *Lachancea fermentati*

Norhayati Yaacob[1,2], Mohd Shukuri Mohamad Ali[1,2], Abu Bakar Salleh[1,2] and Nor Aini Abdul Rahman[3]

[1] Department of Biochemistry, Universiti Putra Malaysia, Malaysia
[2] Enzyme and Microbial Technology Research Centre, Universiti Putra Malaysia, Serdang, Malaysia
[3] Department of Bioprocess Technology, Universiti Putra Malaysia, Malaysia

Corresponding author
Mohd Shukuri Mohamad Ali,
mshukuri@upm.edu.my

## ABSTRACT

**Background.** Not all yeast alcohol dehydrogenase 2 (ADH2) are repressed by glucose, as reported in *Saccharomyces cerevisiae*. *Pichia stipitis* ADH2 is regulated by oxygen instead of glucose, whereas *Kluyveromyces marxianus* ADH2 is regulated by neither glucose nor ethanol. For this reason, ADH2 regulation of yeasts may be species dependent, leading to a different type of expression and fermentation efficiency. *Lachancea fermentati* is a highly efficient ethanol producer, fast-growing cells and adapted to fermentation-related stresses such as ethanol and organic acid, but the metabolic information regarding the regulation of glucose and ethanol production is still lacking.

**Methods.** Our investigation started with the stimulation of ADH2 activity from *S. cerevisiae* and *L. fermentati* by glucose and ethanol induction in a glucose-repressed medium. The study also embarked on the retrospective analysis of ADH2 genomic and protein level through direct sequencing and sites identification. Based on the sequence generated, we demonstrated ADH2 gene expression highlighting the conserved NAD(P)-binding domain in the context of glucose fermentation and ethanol production.

**Results.** An increase of ADH2 activity was observed in starved *L. fermentati* (LfeADH2) and *S. cerevisiae* (SceADH2) in response to 2% (w/v) glucose induction. These suggest that in the presence of glucose, ADH2 activity was activated instead of being repressed. An induction of 0.5% (v/v) ethanol also increased LfeADH2 activity, promoting ethanol resistance, whereas accumulating acetic acid at a later stage of fermentation stimulated ADH2 activity and enhanced glucose consumption rates. The lack in upper stream activating sequence (UAS) and TATA elements hindered the possibility of Adr1 binding to LfeADH2. Transcription factors such as SP1 and RAP1 observed in LfeADH2 sequence have been implicated in the regulation of many genes including ADH2. In glucose fermentation, *L. fermentati* exhibited a bell-shaped ADH2 expression, showing the highest expression when glucose was depleted and ethanol-acetic acid was increased. Meanwhile, *S. cerevisiae* showed a constitutive ADH2 expression throughout the fermentation process.

**Discussion.** ADH2 expression in *L. fermentati* may be subjected to changes in the presence of non-fermentative carbon source. The nucleotide sequence showed that ADH2 transcription could be influenced by other transcription genes of glycolysis oriented due to the lack of specific activation sites for Adr1. Our study suggests that if Adr1 is not capable of promoting LfeADH2 activation, the transcription can

be controlled by Rap1 and Sp1 due to their inherent roles. Therefore in future, it is interesting to observe ADH2 gene being highly regulated by these potential transcription factors and functioned as a promoter for yeast under high volume of ethanol and organic acids.

## INTRODUCTION

*Lachancea sp*, a type of yeast, is no stranger to the study of alcohol dehydrogenase (ADH) gene isolations and biochemical characterizations. A decade ago, ethanol metabolism in *Lachancea sp.* has been studied by *Duarte et al. (2004)* via enzyme profiling showing the differences in ADH isoenzymes which could be observed across *Lachancea* species. ADH catalyzes the final metabolic step in ethanol fermentation and plays an important role for general metabolic pathways of yeast to yield energy as well as to proliferate under anaerobic conditions (*Liang et al., 2014*).

*L. fermentati* was a former member of *Kluyveromyces, Saccharomyces* and *Zygosaccharomyces* (*Kurtzman, 2003*). The identification of *L. fermentati* was based on the evaluation from the perspective of the multigene sequence analysis which resulted in the reassignment of *Lachancea sp.* The haploid number of chromosomes in *Lachancea sp.* is reported to be 8, equivalent to half of the haploid chromosome numbers of *Saccharomyces* and *Kluyveromyces* (*Naumova, Serpova & Naumov, 2007*).

To date, *L. fermentati* is known for its efficient role in ethanol fermentation (*Natarajan et al., 2012*). However, there is a lack of information regarding the metabolic and gene regulations that took place during the glucose fermentation of *Lachancea sp.* (*Thomson et al., 2013*). In this study, ethanol, glucose, and acetic acid are investigated for their role in regulating *L. fermentati* ADH2 expression. This enzyme would then be compared to the expression of common yeast, *S. cerevisiae* ADH2 in terms of repression and derepression mechanism in relation to carbon sources. Glucose repression and derepression essentially concern genes involved in oxidative metabolism (*Weinhandl et al., 2014*). In fermentation, yeast cells accumulate fermentation products, such as ethanol, from sugars. Then, this was accompanied by an increase in medium acidity, due to the secretion of organic acids at the later stage of fermentation. The pathway associated with stress response in yeast, in line with ethanol and acetic acid production, proved to be deadly, as they demonstrate inhibition of cell growth and represses glucose transport (*Burtner et al., 2009*). Because ADH2 gene is highly deregulated by glucose, it can control both of the respiration and metabolism function through its role in aerobic respiration and mitochondrial function. This makes the gene one of the suitable targets for its ethanol and organic acid tolerance. As for this reason, ADH2 can become an important marker in determining the beginning of glucose, ethanol and acetic acid stress activity (*Denis, Ferguson & Young, 1983*; *De Smidt, Du Preez & Albertyn, 2008*; *Lin et al., 2010*).

The gene can also be linked to the function of hexokinase and other glycolytic genes as it has been demonstrated having a promoter activity based on the glucose stimulation (*Lee & DaSilva, 2005*; *Weinhandl et al., 2014*). The long-term advantage of utilizing ADH2 gene as a promoter compared to the other common used promoters is that no specific inducer is required. In the case when an increase of biomass concentration was evident, the ADH2 gene has shown a high level of expression which is appropriate in the optimization of bioethanol fermentation (*Weinhandl et al., 2014*).

The mechanism of yeast stress tolerance was extensively studied in *S. cerevisiae* as compared to other yeasts. It was reported that *S. cerevisiae* could tolerate various types of stresses in particular of ethanol inhibition and osmotic pressure from acids and sugars from bioethanol production (*Zhao & Bai, 2009*). The latest development of stress tolerance mechanisms was focused more on the genomics rather than proteomics as it quick to point out the causes of these adverse conditions. However, the mechanism of yeast stress tolerance through correlation of biological response with the flux of gene expression under the presence of stress is equally important as it elaborates the genetic dependence of stress on the external stimuli of fermentation. Hence, by comparing the expression of ADH2 before and after the induction of these stresses, the role of ADH2 can be evaluated. Besides the observation of genes expression to demonstrate environmental liability that potentially leads to the early sign of stress, it is also vital to know the habitual of genes expressed under a favorable condition for example when glucose is in abundance.

Despite these well–known characteristics, *De Smidt, Du Preez & Albertyn (2012)* reported that ADH2 was not fully repressed even in the presence of glucose, as the gene transcription could still be detected in glucose-laden condition. This could be due to the upstream activation sequences (UAS1 and UAS2) located at the C-terminal of ADH2 sequences. Both of them are necessary for the complete derepression of ADH2 which requires activation by trans-acting regulatory element, Adr1p (*De Smidt, Du Preez & Albertyn, 2008*). For UAS2, the activation (derepression) of ADH2 expression is highly dependent on its orientation and copy numbers. However, when UAS2 is disorientated, a decline of ADH2 expression will be observed as it acts synergistically with UAS1 as a binding site for Adr1p to stimulate the expression of ADH2 (*De Smidt, Du Preez & Albertyn, 2008*; *De Smidt, Du Preez & Albertyn, 2012*). Thus, the uses of ADH2 as a promoter for bioethanol production could be based on its characteristics in the repression and derepression mechanism (*Donoviel, Kacherovsky & Young, 1995*; *De Smidt, Du Preez & Albertyn, 2008*). The choice of the right promoter is a crucial point for efficient gene expression, as most regulations take place at the transcriptional level (*Weinhandl et al., 2014*).

An over-expression of ADH2 is capable of exhibiting a continuous cell growth in ethanol and is important for the control of the glucose uptake. The increase of ADH2 activity also results in the increase of acetic acid production which negatively affects the fermentation, leading to a sluggish or arrested fermentation (*Maestre et al., 2008*). As ADH2 expression increased (derepression), the accumulation of ethanol within the cell is likely to be responsible for the rapid fermentation at 30 °C, influencing high cell density and acetic acid production (*Nagodawithana & Steinkraus, 1976*). The transcription of yeast ADH2 in a regulation of the external environment of fermentation is a very

interesting subject, as it supports in the description of the mechanism involved in various biotechnological processes (*Cho & Jeffries 1999*; *Lin et al., 2010*). In this study, we analyzed the sequence harboring the ADH2 gene from *L. fermentati*, identify its regulatory genes and transcription binding sites, followed by determination of ADH2 regulation in the presence of glucose, ethanol, and acetic acid.

## MATERIALS AND METHODS

### Yeast strains validation and culture conditions

*S. cerevisiae* obtained from a commercially produced yeast powder was used as a control in this study as it demonstrates ethanol fermentation ability in high glucose content (10%, v/v). This is an important criterion in order to observe the expression or activity of a glucose-regulated ADH2 gene. *L. fermentati* was isolated from a fermented nypa sap of *Nypa fruticans* located in Telok Intan, Malaysia (*Natarajan et al., 2012*). The Internal transcribed spacer (ITS) regions (comprising partial sequence 18S rDNA-ITS1-5.8S rDNA-ITS2- partial sequence 28S rDNA) of the selected yeasts were amplified and sequenced for species and strain identification purposes using the methods of *Fujita et al. (2001)*. Based on the BLAST analysis, the ITS nucleotide sequences of *S. cerevisiae* exhibited 98–99% similarity with the *S. cerevisiae* of *Chicha* strain (KC183723.1). As expected, the *L. fermentati* ITS sequence exhibited a 100% similarity with the strain SHM1 (accession number JN674449.1). Yeast Peptone Dextrose (YPD) medium (1% yeast extract, 2% peptone, and 2% glucose) was used for the culture maintenance at 30 °C.

### Partial ADH2 amplification and phylogenetic analysis

Both *L. fermentati* and *S. cerevisiae* were subjected to partial ADH2 PCR amplification by using denatured primer (Table 1). The ADH2 gene from *L. fermentati* strain SHM1 was designed based on the validated alcohol dehydrogenase sequences of *Saccharomyces* and *Lachancea* origins highlighting the conserved region. The amplification procedure was performed in 20 µl reaction with pre-denaturation of 95 °C; 5 min, followed by 30 cycles of denaturation 95 °C; 30 s, annealing 55 °C; 30 s, pre-elongation 72 °C; 30 s, and final elongation 72 °C; for 5 min. The 912 bp length of ADH2 gene obtained from wild-type *L. fermentati* were excised and purified using QIAquick Gel Extraction Kit (QIAGEN, Germany), before submitted for DNA sequencing (1st Base Sdn Bhd, Malaysia). The deposited nucleotide sequences (accession number: KU203771) were then translated into peptide sequences using a translational tool from ExPASy (SIB, Switzerland) and peptide alignments were performed by using SDSC Biology Workbench 3.2. Phylogenetic analysis based on the model of Pearson and the neighbor-joining analysis was performed with 1,000 replicates of bootstrap test by using MEGA 6.06 software.

### Total RNA extraction and cDNA synthesis

Total RNA was extracted using the Presto$^{TM}$ Mini RNA Yeast Kit (Geneaid, Taiwan) following the protocol provided by the manufacturer. RNA concentration and quality were determined using the $A_{260}/A_{280}$ spectrophotometric absorption ratio (Eppendorf, Germany) and RNA Bioanalyzer 2100 Plant Nano chip (Agilent, USA) to determine the

**Table 1** List of the primers for ADH2 gene amplifications and one reference gene for the normalization of the gene expression studies.

| No. | Primer | Sequence (5′-3′) | Start (bp) | Stop (bp) | Tm (°C) | Length (bp) |
|---|---|---|---|---|---|---|
| 1. Degenerate primer amplification based on *Lachancea kluyveri* ADH2 (accession no. AAP51047.1) gene sequence for unknown ADH2 gene isolation from *L. fermentati* strain SHM1 (to provide sequence template for selected gene targets) | | | | | | |
| | ForADH2 | ATYCCAGAAACTCAAAARGCCRTTA | 538 | 565 | 52.5 | 912 |
| | RevADH2 | TGTCRACAACGTATCTACCRRCAAT | 1,547 | 1,572 | 59.3 | |
| 2. Real time RT-PCR primer sequences (derived from sequences from partial amplified ADH of respective strain) | | | | | | |
| i | SceADH2_F | CGCAGTCGTTAAGGCTACCAA | 690 | 710 | 59 | 70 |
| | SceADH2_R | CGATAGCGGCTTCGGAAAC | 760 | 742 | 59 | |
| ii | LfeADH2_F | GACTTTACCAAGACCAAGG | 655 | 673 | 62 | 61 |
| | LfeADH2_R | ACCTTGAGCACCACCGTTGGTG | 720 | 699 | 66.3 | |
| iii | ACT1_F* | TGGATTCCGGTGATGGTGTT | | | 56 | 72 |
| | ACT1_R* | TCAAAATGGCGTGAGGTAGAGA | | | 56 | |

**Notes.**
ACT1* was selected as the reference gene for normalization.

RNA integration number (RIN) values. The RNA quality was also tested by electrophoresis on 1% (w/v) agarose gel. cDNA was synthesized from the isolated high-quality RNA by using one-step RT-qPCR iScript$^{TM}$ reverse transcription supermix (BioRad, USA).

## Real time-PCR primer design

A complete nucleotide sequence of *Saccharomyces cerevisiae* ADH2 was available at the *Saccharomyces cerevisiae* Genome Database (http://www.yeastgenome.org/). The primers for *S. cerevisiae* were designed by using the software Primer Express (Applied Biosystems, Foster City, USA), while the primers for *L. fermentati* were designed based on the multi-alignment of ADH2 gene sequences from the *Saccharomyces*, *Kluyveromyces*, and *Lachancea* families. Actin1 (ACT1) was chosen as the reference gene for normalizations (*Ismail et al., 2013*). Real-time PCR Table 1 showed the primer sequences used for gene expression along with their amplicon sizes.

## Gene expression analysis with real-time PCR

ADH2 gene expression was determined by real-time PCR. An efficiency test was performed using cDNA from *S. cerevisiae* and *L. fermentati* as the template. The PCR efficiency was scaled to 95–105% and above, 0.999 R-squared and a slope of −3.33, as recommended (*Bustin et al., 2009*). Real-time PCR assays were performed in an iCycler (iQ$^{TM}$5, BioRad, USA), using the standard thermal cycling protocol. The reaction mixture of 20 µl contained 10 µl of iTaq Universal SYBR$^{®}$ Green Supermix (2x) (BioRad, USA), 0.75 nM of forward and reverse primers, 2 µl of 10-fold diluted cDNA, and deionized water to reach the final volume. Real-time PCR experiments were carried out in two replicates, of which each real-time PCR was performed in duplicate on each sample. A negative control without the cDNA template was included. The thermal amplification program used was as follows: 95 °C for 30 s; 40 cycles of 95 °C for 10 s; and 60 °C for 30 s before the melt-curve was collected from 58 °C to 90 °C. Figure S1 displayed melt curve analysis for ADH2 and Act1

genes during the amplification. Gene expression levels were shown as the threshold cycle ($C_t$) of the studied gene normalized with the $C_t$ of a reference gene, ACT1 (Table S1).

## Induction of ADH2

In ADH2 protein induction study, 2% (wv$^{-1}$) glucose (for repression) and 0.5% (vv$^{-1}$) ethanol (for derepression) were added in 100 ml of YPD medium at 30 °C with 200 rpm of agitation after 24 h because there was no glucose remained in the medium. Following the induction, fermentation was extended for one hour before cells were completely harvested and protein was extracted for ADH2 activity determination.

## Intracellular protein extraction

The cell was harvested from the culture and washed two times with deionized water. Then, the protein extract was prepared by mixing 20 ml of 50 mM phosphate buffer (pH 7.8). The cell suspension was lysed via the sonication method (Branson, USA) for 2.5 min, with 5 s of pulses. The supernatant was collected as protein extract after 10 min of centrifugation at 8,000 × g with a temperature of 4 °C. Total protein concentration was determined by the method of Bradford (Quick Start$^{TM}$ Bradford, BioRad), as specified by the manufacturer.

## ADH2 screening assay

The unpurified soluble protein extracted from *S. cerevisiae* and *L. fermentati* cells were used in the measurement of ADH2 activity. The reaction for ADH2 assay was performed in 0.3 ml of total volume containing 20 mM of sodium pyrophosphate buffer (pH 7.8), 3.3% (v/v) of ethanol, 7.5 mM of NAD$^+$ and <10 µg of ADH (protein), with 7–8 min of reaction time. ADH2 activity was measured by increasing 340 nm absorbance in response to NADH or acetaldehyde production. The units/ml of ADH2 activity was calculated by using 6.22 of the millimolar extinction coefficient of $\beta$-NADH at 340 nm. One unit of NAD$^+$ converts 1.0 µmole of ethanol to acetaldehyde per minute at pH 7.8 and 25 °C (*Kägi & Vallee, 1960*). ADH activity was also qualitatively measured via native PAGE by adding ADH activity buffer containing 4.0 mg of phenazine methyl sulfate (Sigma-Aldrich, Germany), 10 mg of nitroblue tetrazolium (Sigma-Aldrich, Germany), 50 mg of NAD$^+$ (Sigma, Germany) and 0.05 ml of absolute ethanol dissolved in 50 ml of 0.1 M Tris–HCl buffer (pH 8.5) until the desired purple staining was formed (*Young et al., 2000*).

## Shake flask fermentation

Seed cultures of *S. cerevisiae* and *L. fermentati* were grown at 30 °C, 200 rpm agitation speed for 24 h in a universal tube containing 10 ml of the YPD medium and then inoculated into the fermentation medium at a level of 1% (vv$^{-1}$). The fermentation medium of 2% YPD (1% yeast extract, 2% peptone, and 2% glucose) was prepared at 100 ml in a 500 ml Erlenmeyer flask, where the fermentations were all carried out in similar condition. Before the fermentation, the starting density (OD$_{600}$) of all cultures was 0.5. Over the course of the 30 h fermentation, samples were collected every 6 h to monitor the growth, ethanol, organic acid and residual glucose concentrations. Repeats were performed in triplicate. The fermentation products were centrifuged at 8,000 × g for 10 min to separate the cells from the analytes. The supernatants were stored at −20 °C until used for the ethanol and glucose assays while the precipitants were used in the RNA extraction and ADH assays.

### Determination of ethanol and glucose concentration

The content of the ethanol in the fermentation samples was determined based on the 96-microwell plate system using the ethanol extraction method (*Seo et al., 2009*). The sugar concentration was determined using the 3, 5-dinitrosalicylic acid (DNS) method (*Bernfeld, 1955*).

### Acetic acid production determination

Analyzes of organic acids, such as ethanoic acid (acetic acid) were performed by High-Performance Liquid Chromatography, using an Agilent Poroshell 120 EC-C18 column. The column and mobile phase line of the instrument were thoroughly washed with 50% Acetonitrile/50% water of mobile solution. The running buffer or eluent that was used to detect weak acids contained 1 mmol/l of sulfuric acid and 8 mmol/l of $Na_2SO_4$, with pH 2.8. The flow rate of the mobile phase was adjusted to 0.9 ml/min and the temperature was fixed at 25 °C.

## RESULTS AND DISCUSSION

### Derepression of ADH2 by glucose induction

While the era of yeast synthetic biology began in the well-characterized model organism *S. cerevisiae*, other non-conventional yeast production systems such as *Hansenula polymorpha*, *Kluyveromyces lactis*, *Pichia pastoris*, and *Yarrowia lipolytica* have been regarded as a eukaryote model with growing importance (*Wagner & Alper, 2015*). These yeasts including *L. fermentati* have roles in the manufacture of vaccines, therapeutic proteins, food additives, and renewable chemicals, but recent synthetic biology advances have the potential to greatly expand and diversify their impact on biotechnology, which is how the potential of ADH2 gene from *L. fermentati* (LfeADH2) can provide an alternative biomanufacturing platforms primarily as a promoter. To date, *L. fermentati* has not yet been widely considered for industrial usages.

During induction at the glucose-limiting period, ADH2 from *L. fermentati* exhibited higher affinity for glucose and ethanol, as shown by a 2.5-fold increase in 0.5% ($vv^{-1}$) ethanol and 0.5-fold increase in 2% ($wv^{-1}$) glucose (Fig. 1). However, *S. cerevisiae* ADH2 activity was increased by 1.3-fold following the addition of 2% glucose, and was repressed by 0.5-fold with the addition of 0.5% ethanol. Based on the results obtained, ADH2 protein from *Saccharomyces cerevisiae* did not exhibit a strong glucose-repressible activity.

ADH2 activity staining using the ADH2 protein confirmed the outcomes of ADH2 activity after the sugar and non-sugar-based induction. The degradation of ethanol as a substrate by this protein-entrapped in native polyacrylamide gel can be measured by its intensity after staining. Based on these results, it can be implied that both glucose and ethanol shared significant roles in the derepression or activation mechanism of *L. fermentati* ADH2 promoter.

This experiment demonstrated that both ADH2 enzyme activity and protein expression shifted with the addition of ethanol or glucose into the medium. In previous work by *Irani, Taylor & Young (1987)*, ADH2 from *S. cerevisiae* was undetectable in the presence of a fermentable carbon source such as glucose, but when glucose was exhausted from
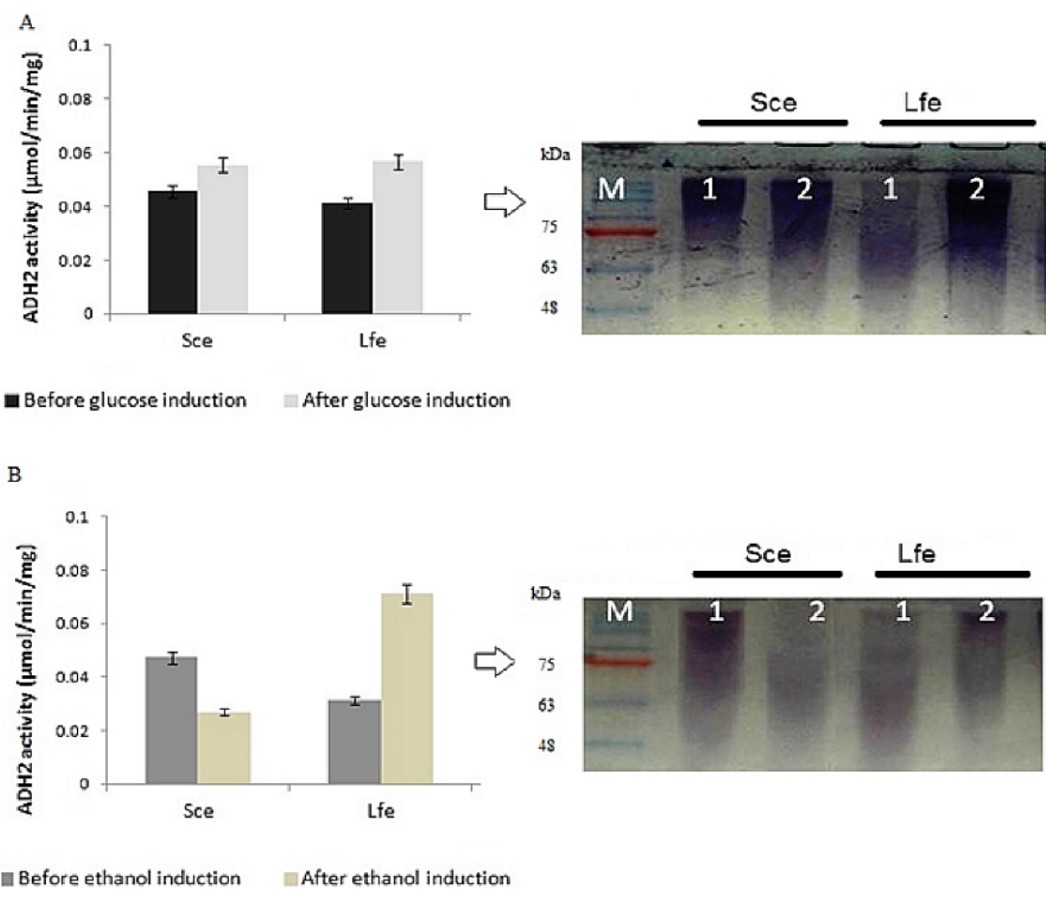

**Figure 1** **Glucose and ethanol induction on ADH2 from *S. cerevisiae* and *L. fermentati*.** ADH2 activity in *S. cerevisiae* (Sce) and *L. fermentati* (Lfe) after induction with 2% (wv$^{-1}$) glucose (A) and 0.5% (vv$^{-1}$) ethanol (B). Qualitatively, the ADH2 activity staining was performed on the non-denaturing polyacrylamide gels with respective lanes indicated as 1 (before induction) and 2 (after induction). All experiments were repeated twice for validation.

the medium or when the cells are grown on a non-fermentable carbon source such as ethanol, ADH2 enzyme activity, and mRNA are present in large amounts. The results were connected to *Denis, Ferguson & Young (1983)*, which described contradicting changes in *S. cerevisiae* ADH1 expression resulting an increase of mRNA levels when ethanol grown-cells were transferred into medium containing glucose (*Tornow & Santungelo, 1990*). Therefore, it is not surprising that both ADH1 and ADH2 have been widely accepted as a promoter in yeast for their regulation according to a specific carbon source. In this study, ADH2 gene from *L. fermentati* strain SHM1 has been found to be regulated by both glucose and ethanol, an interesting concept of a two-in-one promoter system.

Previously, *S. cerevisiae* ADH2 promoter is not the only alcohol dehydrogenase promoter that has been used in the study of glucose-related expression. Fission yeast *S. pombe* ADH2 promoter which showed high homology at the protein level to *S. cerevisiae* ADH2 also was frequently utilized as a promoter. Unlike *S. cerevisiae*, *S. pombe* ADH2 was constitutively expressed that resulted in the up-regulation of ADH2 transcription in the presence of

glucose (*Russell & Hall, 1983*). Contradict to that, *K. marxianus* strain IFO 1802 ADH2 (KmADH2) exhibited a low expression level in high glucose and ethanol throughout the 72 h of 150 g/L glucose fermentation (*Liang et al., 2014*). Overall, the stringency of yeast ADH2 activation and expression towards the glucose availability was rather obscure. There are mutants that allowed ADH2 expression to escape glucose-repression in which, a rare-semidominant Adr1 allele (constitutive) is involved. The constitutive Adr1 showed mutations between amino acids 227 and 239 was said to be caused by a post-translational inhibition (*Ratnakumar et al., 2009*). Inclusive to our findings, the glucose induction reflects the possibility of ADH2 to be express in the presence of glucose. It means that glucose and ethanol would have the ability to promote ADH2 expression in *L. fermentati*. This result was contradicted to *K. marxianus* ADH2 where the protein was unable to be expressed in the presence of either of this compound.

Based on these findings, we performed some retrospective analysis on sequences bearing *L. fermentati* ADH2-like sequences highlighting on the presence of transcription binding sites which probably involved in the activation of ADH2 transcription.

## Multiple alignments of partial ADH peptide sequences

The partial ADH gene amplifications were initiated by using a pair of denatured primer (Table 1). The primer sequences were designed based on the multi-alignment of verified ADH2 nucleotide sequences from *Saccharomyces* and *Lachancea* species. To date, there are no alcohol dehydrogenase sequences available in NCBI database for *L. fermentati*. Thus, we amplified and sequenced ADH2 gene from *L. fermentati* (previously known as *Zygosaccharomyces fermentati*) for the first time. The 912 bp-length ADH2 nucleotide sequences amplified from *L. fermentati* was very different from the well-known yeast ADH2 sequences. The highest similarity score of ADH2 sequences amplified from *L. fermentati* strain SHM1 (KU203771) was 88.6% with *L. kluyveri* ADH2 (AAP51047.1). The nucleotide sequences and motifs observed in ADH2 gene in *L. fermentati* were highly conserved. But the analysis showed that there were a few variations found to the other existing ADH2 sequences from different species. Figure 2 showed the multiple alignments of various ADH2 protein sequences together with the ADH2 peptide sequence from *L. fermentati*. The analysis of ADH2 sequences from *L. fermentati* successfully identified the highly preserved amino acids and the zinc-binding motifs. Based on the protein alignments, several residues were believed to have a species-related function as observed in Iso-37, Glu-117, Glu-175, Cys-183, Thr-228, Val-231 and Gly-276. It is, therefore, important to study on the role of these residues toward the expression of this gene as they could provide be a useful indication on the nature of regulation of this protein. In this assignment, Thr-228 and Val-231 were the residues included in sequences for gene expression throughout the yeast fermentation study. Typically the HXXXH motifs containing imidazole ring has been well described in yeast as the site where metal ion exchange happens in zinc-containing enzymes. Currently, both *S. cerevisiae* and *L. fermentati* exhibited conservation of this zinc binding sites at residues 48–55 based on the HXXXH motif and residues 64–74 according to the Interpro analysis (*Obradors et al., 1998*).

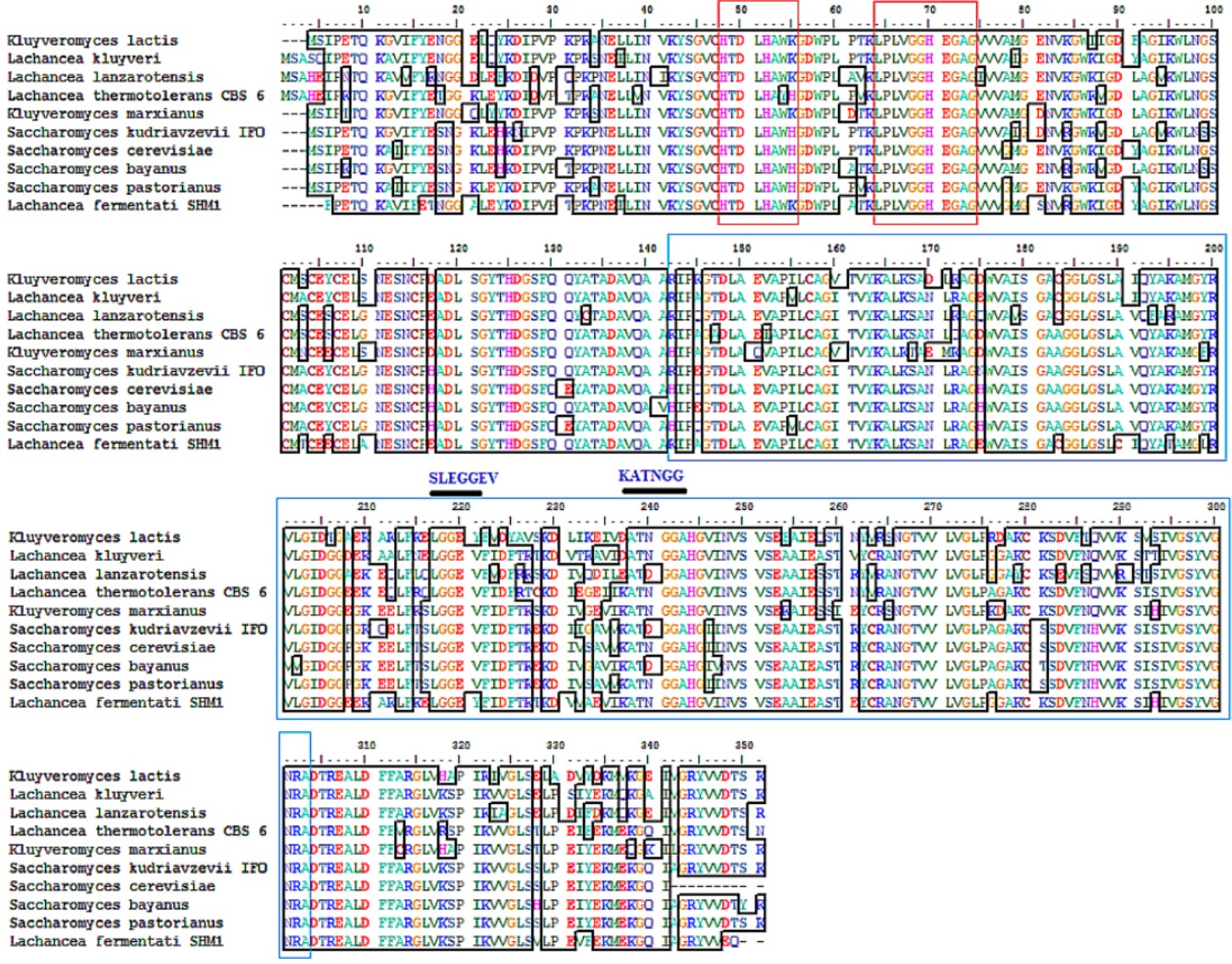

**Figure 2** **Multiple ClustalW alignment of ADH2 peptide sequences.** The red box showed amino acid alignment of zinc-binding site (HXXXH) present in the medium-chain NAD (P)-dependent dehydrogenase. While the blue-box contained the sequences of NAD (P)-binding domain having the motifs of SLEGGEV and KATNGG used for gene expression. The N-terminal of ADH2 gene from *Lachancea fermentati* strain SHM1 starts at residue 29–137, while the C-terminal starts at residue 180–304 based on Interpro (EMBL-EBI) analysis.

The neighbor-joining phylogenetic analysis of ADH2 nucleotide sequences from *L. fermentati* strain SHM1 exhibited highest similarity to ADH2 gene belonged to *L. kluyveri* followed by the *Saccharomyces* family. A slight distance from *L. fermentati* observed in the phylogenetic tree indicated the distribution of *Kluyveromyces* and another different *Lachancea* cluster family, where this isolated cluster of *Lachancea* group contained the uncharacterized ADH2 sequence. Hence, *L. fermentati* ADH2 was presumed to inherit considerable genomic characteristics from the *Saccharomyces* and *Kluyveromyces* origins (Fig. 3).

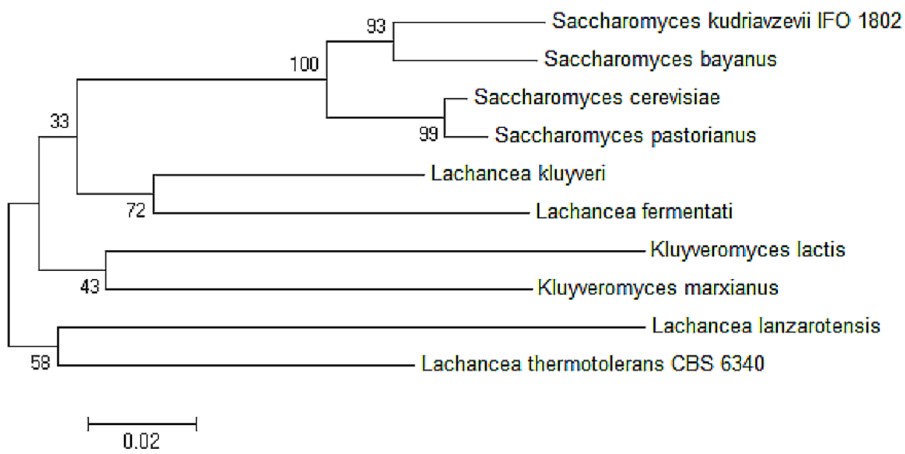

**Figure 3** Neighbor-joining phylogenetic analysis separating the sequences of ADH2 from *L. fermentati* in *Lachancea* ADH2 family and *Saccharomyces/Kluyveromyces* ADH2 family groups.

## The transcription binding site in partial ADH2 sequences of *L. fermentati*

Yeast ADH2 was known to repress growth on fermentative carbon sources and have a lower affinity for glucose (*Kachevrovsky et al., 2008*). The results of glucose-inducible ADH2 contradicted the normal regulation of glucose-repressible ADH2, as observed in real-time PCR, indicating that ADH2 transcription was not glucose-dependent. To date, it is known that the glucose-regulated ADH2 gene was activated by several regulating genes. Among these genes is Adr1, which binds to the ADH2 promoter site at upper-activating sequence (UAS). When glucose was added, the established Adr1-UAS1 binding weakened, as Adr1 was inhibited by glucose, which, in turn, repressed ADH2, causing minimal expression or activity (*Irani, Taylor & Young, 1987*; *Young et al., 2000*). However, *Young et al. (2008)* reported that Adr1 competently binds to the specific UAS sequence even when glucose levels are still high, which may be a possible reason for the high ADH2 activity observed after glucose was added. Alternatively, the increase in Adr1 synthesis could also cause ADH2 to express constitutively during growth on glucose. This condition allows ADH2 expression to escape glucose repression (*Irani, Taylor & Young, 1987*). However so, this type of regulation had only been studied in *S. cerevisiae* and has not been evaluated for other kinds of yeast species. Figure 4 demonstrates that the conservation of Adr1-binding site was not always significant for glucose-regulated genes as most of the ADH2 sequences from yeast which belonged to non-*Saccharomyces* group possessed no inherent Adr1-binding sites. A multiple sequence alignment of ADH2 proteins from *P. stipitis* (Y13397.1), *S. cerevisiae* S288c (NM_001182812.1), *K. marxianus* (KF678866.1) and *L. fermentati* (KU203771) showed some irregularities in the conservation sites of UAS and TATA sequences (Fig. 4). Nucleotide sequence analysis showed that *P. stipitis* and *S. cerevisiae* possessed UAS and TATA repeats while *K. marxianus* and *L. fermentati* did not.

In *S. cerevisiae*, the presence of large excess of competing promoter elements such as UAS and TATA regions were important to ensure the competency of ADH2 promoter
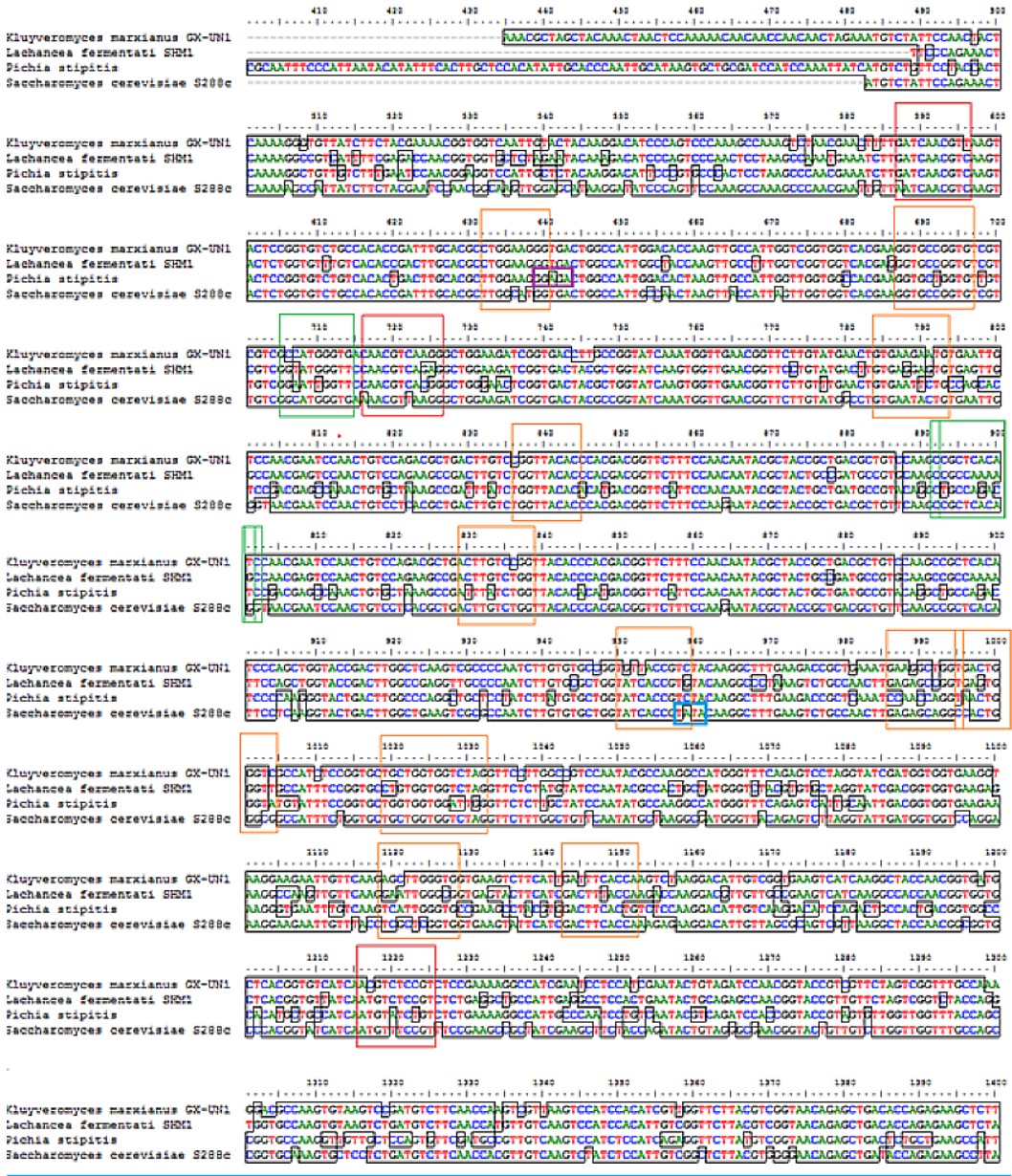

**Figure 4** Multiple sequence alignment of *Pichia stipitis* (Y13397.1), *Saccharomyces cerevisiae* (NM_001182812.1), *Kluyveromyces marxianus* (KF678866.1) and *Lachancea fermentati* (KU203771). The presence of UAS sequences (5′-GGAGA-3′) was identified in a purple box and TATA elements were captured in blue box. Three transcription genes were identified as Sp1 (11) in the orange box, Rap1 (3) in the red and Adr1 (3) represented in the green box.

(*Irani, Taylor & Young, 1987*). The involvement of Adr1 transcription factor to promote ADH2 expression was primarily based on the conservation of UAS1 and UAS2 regions. The ADH2 promoter which lacked the upstream regulatory region has been reported to cause in inactivation, while the missing TATA box in the sequence was reported to contribute to an intermediate level of binding competition (*Irani, Taylor & Young, 1987*). It is worth to note that the control of Adr1 binding does not only occur at the gene transcriptional level but also during the post-translational modification of the protein as

well (*Irani, Taylor & Young, 1987*; *Kachevrovsky et al., 2008*). Following the preliminary genomic studies, ADH2 gene transcription in *L. fermentati* might not work according to its approach (glucose-repressible) in the absence of UAS and TATA.

The presence of Adr1-binding sequence at nucleotides 99–108, 228–237, 828–839 showed that ADH2 gene from *L. fermentati* would be readily transcribed. The lack in adjacent upper stream-activating sequences, however, left the sequence incomplete for Adr1 binding. Nonetheless, within the sequence of ADH2, there are other potential regulatory genes which made up the complex transcriptional network in yeast. These genes which exist in abundance can promote ADH2 regulation inadvertently providing suitable condition which stimulates them. In *L. fermentati*, regulatory genes or activator-binding protein such as the zinc finger transcription factor (Sp1) and repressor/activator site binding protein (Rap1) are believed to equally participate in ADH2 gene regulation. Sp1 contains a highly conserved DNA-binding domain composed of three zinc fingers close to the C-terminus with serine/threonine- and glutamine-rich domains in their N-terminal regions. This protein is an extremely versatile protein involved in the expression of many different genes which differs between the cell types during development. Given that there are at least 11 sites for the binding of Sp1 in the *L. fermentati* ADH2 gene and its importance in many gene activations, it is predicted that the cells would not survive without their presence. So far, little is known how Sp1 act on natural promoters in combination with other transcription factors *in vivo* (*Suske, 1999*; *Bouwman & Philipsen, 2002*).

The presence of RAP1 transcriptor factor which was known to activate most ribosomal protein and glycolysis enzymes can function as a strong key promoter to ADH2 gene in the presence of glucose. The Rap1 site appears to mediate the response of these genes to the appearance or disappearance of glucose; for example, their expression is coordinately down-regulated upon depletion of glucose at the diauxic shift (*Santangelo, 2006*). The presence of regulatory genes shown in Fig. 5 was predicted via *in-silico*—based software (TRANSFAC Database, Biobase, GmbH) and was found to be conserved in the nucleotide sequence of ADH2 from *L. fermentati*. Therefore, the ADH2 gene could harbor its function through various promoters and activators without having the UAS and TATA regions to activate it through Adr1.

In general, ADH2 has a broader role in the yeast cell physiology and growth than previously reported, as this gene can either be involved in the utilization of non-fermentable carbon sources, carbohydrate and nitrogen metabolism or protein synthesis depends on its type of species (*Cheng et al., 1994*). In the quest to determine which carbon elements contributes to ADH2 activation in the presence of glucose, ethanol, and by-product (acid), a relative gene expression of ADH2 from *L. fermentati* and *S. cerevisiae* were performed.

## ADH2 gene regulation

In analyzing ADH2 expression, cells from *S. cerevisiae* and *L. fermentati* were harvested at different growth stages. The cell harvest obtained from the basal (zero h), exponential (6th, 12th h), mid-exponential (18th h) and stationary (30th h) phases, along the 30 h length of 2% (w/v) glucose fermentation. The regulations of ADH2 in both yeasts were influenced by cell growth, ethanol production as well as glucose consumption as shown in Fig. 6.

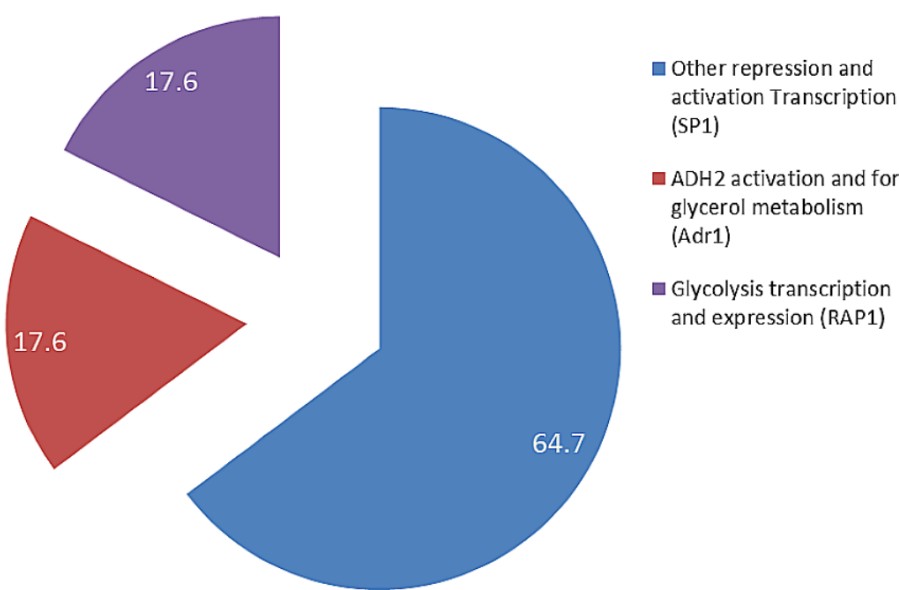

**Figure 5** The fractionation of regulatory transcription genes present in ADH2 gene from *Lachancea fermentati* strain SHM1 (in percentage).

It is true that when glucose gets depleted and ethanol started to build up, the expression of *S. cerevisiae* ADH2 improved showing its activation during exponential growth phase. The trend of ADH2 expression follows the order of consumption and ended with ethanol as well as by-product production. The availability of glucose can influence the repression of ADH2 in the waking phase of fermentation. But once the glucose was completely used up, ethanol or even acetate would take up the role of ADH2 derepression. It has been reported that, at the transcriptional level, the up-regulation of glucose-repressible ADH2 is a result of an increase in ethanol or acetate production (*Ida et al., 2012*; *Weinhandl et al., 2014*).

In LfeADH2, the sluggishness expression in conjunction with the glucose consumption was expected. When ethanol was high, LfeADH2 expression was derepressed. The derepression state of LfeADH2, however, did not last to the end of fermentation, as ADH2 was immediately repressed through the oxidation of ethanol due to its decline or with the presence of other common fermentation by-products such as acetic acid. In this study, *L. fermentati* ADH2 derepression by ethanol was pursued with the increase of ADH2 activity. However, the reason of ADH2 derepression in glucose remained unclear, as glucose was expected to inhibit the transcription of ADH2. One of the reasons would be because of its high rate in glucose consumption that would contribute to the major decline in ADH2 expression rather than the concentration itself. In returns, the expression or activity of ADH2 are more likely to increase when the glucose consumption rate declined and not due to the limitation of glucose in the medium (Table S1).

The outcome of ADH2 gene expression from *L. fermentati* ADH2 showed that the gene has an indispensable function for being tolerant of a higher concentration of ethanol and to maintain cell growth in ethanol. Apparently, the production of ethanol has not improved
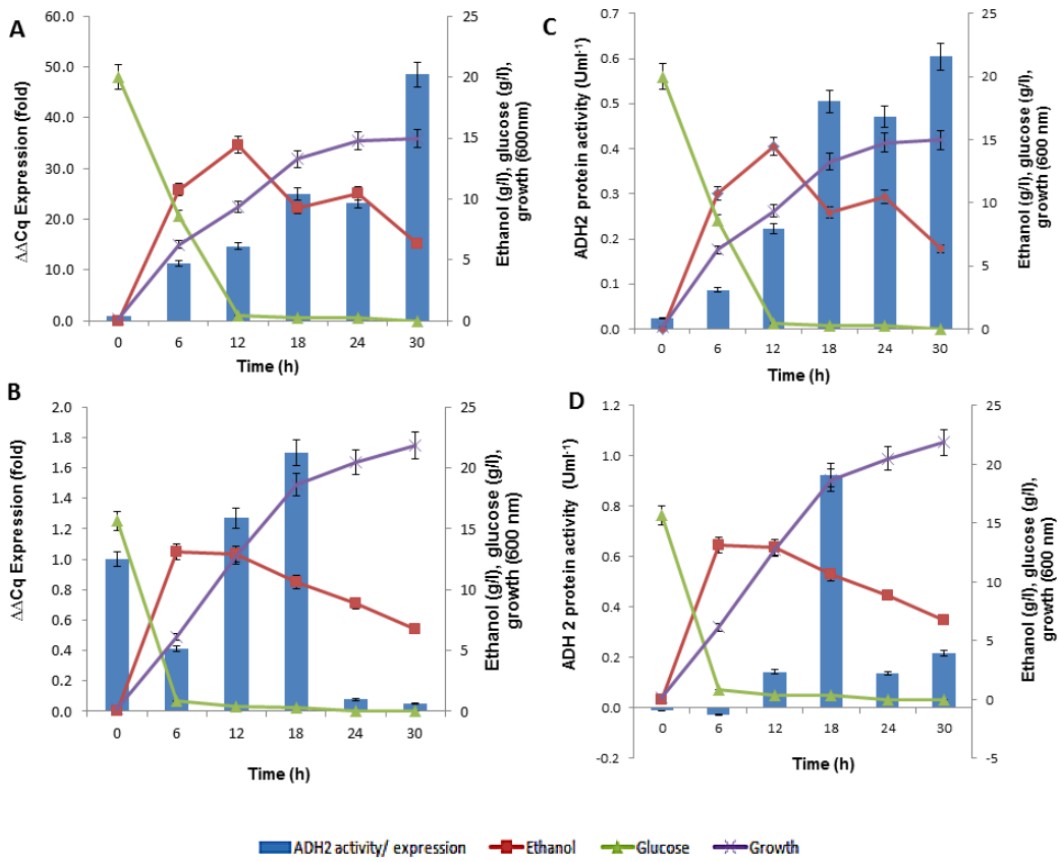

**Figure 6  The profiles of ADH2 gene expression measured by real time PCR and ADH2 activity determination based on protein reaction with ethanol on both *S. cerevisiae* and *L. fermentati*.** (A) and (B) exhibited the relative ADH2 gene expression, whereas (C) and (D) exhibited the ADH2 activity of each protein extracted from *S. cerevisiae* and *L. fermentati* respectively, under similar fermentation state.

following the up-regulations of LfeADH2 expression. The regulation of ADH2 could be influenced by other co-expressed genes, such as pyruvate decarboxylase (PDC). PDC plays a role in acetaldehyde formation during the conversion of pyruvate to acetaldehyde, where it may have led to the reduction of ethanol concentration (*Ida et al., 2012*; *Milanovic et al., 2012*). This would further promote the down-regulation of ADH2 expression. Together, the involvement of PDC and ADH2 were reported to synergistically promote growth in yeasts by glucose metabolism, as these enzymes were capable of efficiently converting intracellular pyruvate and NADH into ethanol. This event typically occurred in a homoethanol fermentation pathway, also known as the Entner-Doudoroff (ED) pathway (*Piriya et al., 2012*). The main difference in the gene expression of ADH2 derived from *S. cerevisiae* and *L. fermentati* has been the result of ethanol. This could be explained by *L. fermentati* ADH2 affinity for ethanol which exhibited lower $K_m$ compared to *S. cerevisiae* ADH2 (Table 2).

In this observation, the activity of ADH2 was identical to the transcriptional profiles of ADH2 exhibited over the fermentation course. The bell-shaped curve of ADH2 activity in *L. fermentati* again indicated that ADH2 transcription was not compelled to the regulation

**Table 2** Enzyme kinetics of crude alcohol dehydrogenase (ADH) from *L. fermentati* and *S. cerevisiae.* All assays were carried out at 25 °C and the absorbance at 340 nm was measured. Assays were performed in triplicates per single run. $V_{max}$ and $K_M$ were calculated based on the half-reciprocal, Hanes-woolf equation (*Ritchie & Prvan, 1996*).

|         | Substrate | $K_M$ (mM) | $V_{max}$ (umol mg$^{-1}$ s$^{-1}$) | $K_{cat}$ (s$^{-1}$) | $K_{cat}/K_M$ (M$^{-1}$ s$^{-1}$) |
|---------|-----------|------------|-------------------------------------|----------------------|-----------------------------------|
| SceADH2 | ethanol   | $0.28 \times 10^2$ | $0.8 \times 10^{-2}$ | $1.18 \times 10^{-2}$ | $42.54 \times 10^{-2}$ |
| LfeADH2 | ethanol   | $5.08 \times 10^2$ | $1.3 \times 10^{-2}$ | $3.86 \times 10^{-2}$ | $7.60 \times 10^{-2}$ |

of glucose-ethanol alone. The increased level of acetic acid might also involve in the regulatory mechanism of ADH2 gene expression, in which the function of this weak acid was known to retain the redox potential under fermentative condition.

## Influence of acetic acid production on ADH2 activity

Acetic acid co-production can influence yeast ADH2 activity during cell growth via the metabolic oxidation of carbon sources that generates large amounts of organic acids. To manage the cytosolic pH homeostasis, the production of acids must be kept in equilibrium with their utilizations (*Zdraljevic et al., 2013*). As ADH2 involved in the oxidation of ethanol and acetic acid production, it was relevant to determine the correlation between the production of organic acid, particularly of acetic acid, and the changes of ADH2 activity. The presence of this acid can contribute to the rate-limiting steps in ethanol fermentation although they are produced to retain intracellular pH in response to glucose availability and to prevent direct inhibition of ethanol via their buffering capacity (*Thomas, Hynes & Ingledew, 2002*; *Ullah et al., 2012*).

Subsequent experiments performed in glucose fermentation also showed that ADH2 activity was very much regulated by acetic acid produced by *L. fermentati* (Fig. 7A). However, *S. cerevisiae* ADH2 activity appeared not to be regulated by acetic acid (Fig. 7B).

According to *Maestre et al. (2008)*, the overexpression of the ADH2 gene resulting in its high activity during alcohol fermentation would affect the glucose uptakes, cell growth and increased production of acetic acid. It was also reported that this condition was associated to detoxification by removal of acetaldehyde (oxidized product) to restore the intracellular redox potential. Judging by the importance of ADH2 expression in *L. fermentati* to prevent cell-induced apoptosis, the role of ADH2 could not be ruled out. Acetic acid can easily penetrate the cell wall and induced cell death and one way to decelerate this process is through promoting ADH2 expression or activity (*Sousa et al., 2012*). Hence, we proposed that acetic acid could be the non-fermentative carbon source, capable of triggering the regulation of LfeADH2 in glucose fermentation as well as to promote cell growth.

ADH2 activity in yeasts come in handy as it has the advantage to promote organic acid and ethanol tolerance and to maintain the relative production of ethanol. In Table S1, ADH2 activity in *L.fermentati* was found high in the presence of acetic acid and ethanol. These concentrations were equivalent to 0.835 mmol/L of acetic acid and 10.62 g/L of ethanol when *L. fermentati* ADH2 activity reached its highest level. Subsequently, ADH2 activities subsided when acetic acid was suppressed. Based on these results, we described that the approach of LfeADH2 in gene transcription was not strictly glucose-regulated,

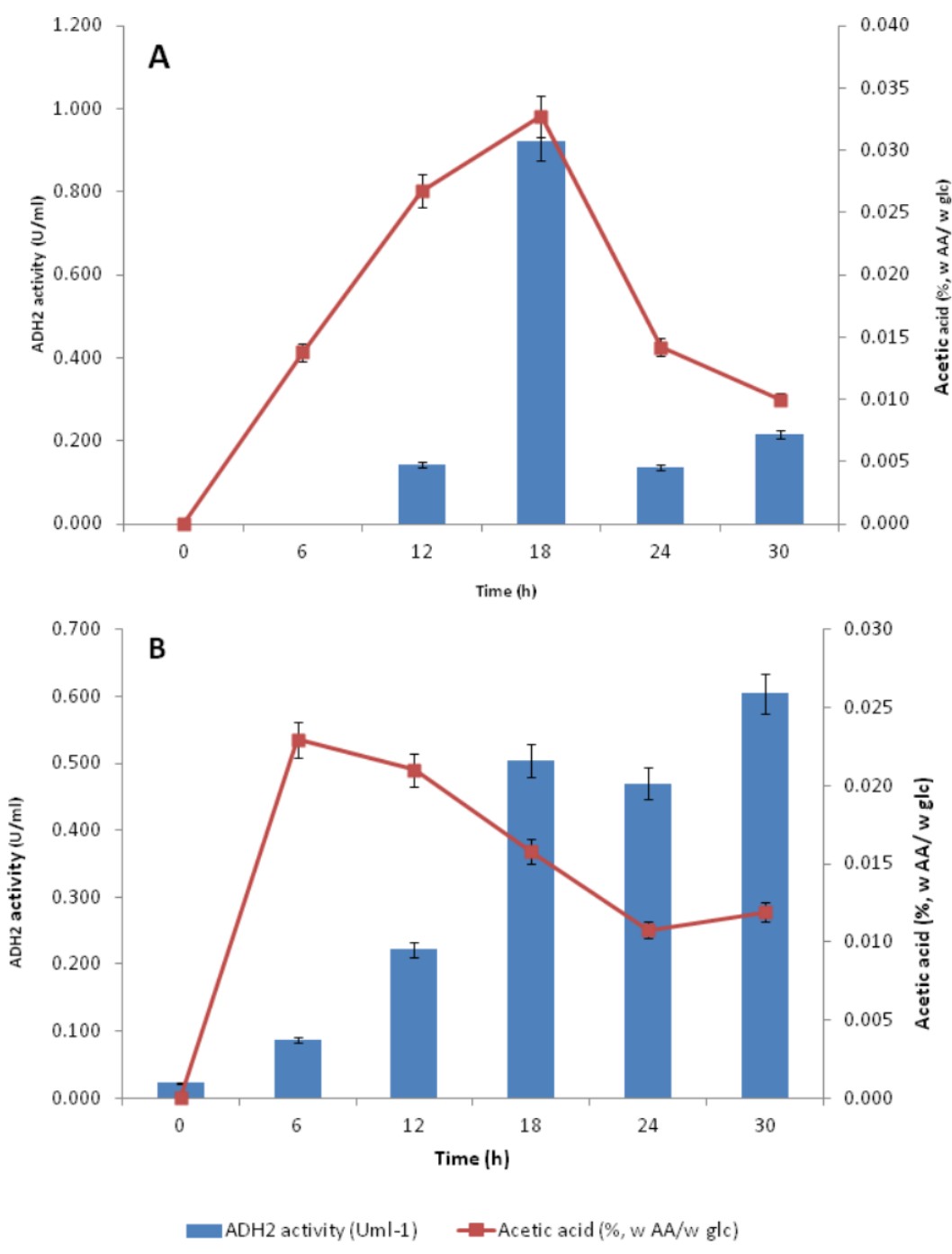

**Figure 7** Effect of acetic acid production on the ADH2 activity from *L. fermentati* (A) and *S. cerevisiae* (B) glucose fermentation.

as specifically observed in previous experiments performed in most of the wild-type *S. cerevisiae* (*Vallari et al., 1992*).

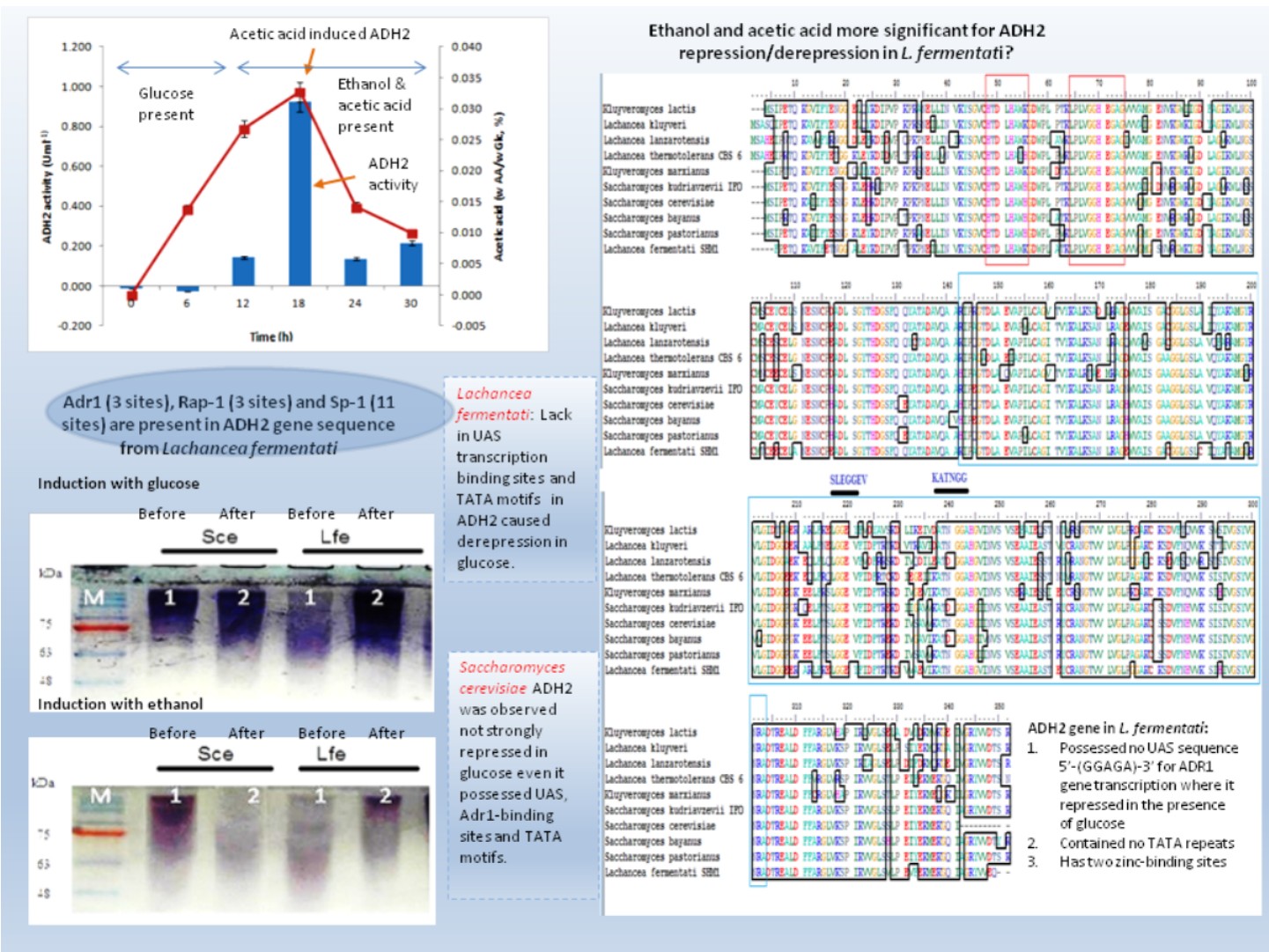

**Figure 8** Schematic diagram describing the role of ADH2 gene from *Lachancea fermentati* strain SHM1 in the regulation of glucose, ethanol and organic acid in bioethanol fermentation.

## CONCLUSION

Many of the cellular and metabolic features of *L. fermentati* ADH2 showed protective roles toward intracellular processes during ethanol fermentation. The up-regulation of *L. fermentati* ADH2 in the presence of glucose and ethanol showed that ADH2 enabled both glucose utilization and ethanol production to occur simultaneously. Acetic acid also is a precursor to ADH2 activity, and *L. fermentati* has showed to demonstrate a higher tolerance to acetic acid prior to the increase in ADH2 activity. Figure 8 showed a schematic diagram of ADH2 sequence from *L. fermentati* which plays an important role to the cause of gene transcription influencing protein expression during glucose fermentation. Generally, ADH2 has become of practical importance for bioethanol production from *L. fermentati*, as this gene was highly regulated at the highest level of ethanol fermentation. The mechanism

of Adr1–ADH2 binding in *L. fermentati* can be unique but the absence of UAS and TATA elements would mean that the gene was unable to be transcribed based on the glucose-repressible condition. Alternatively, the presence of other well-described regulating genes such as Rap1 and Sp1 could be of importance to the activation of *L. fermentati* ADH2. Based on this preliminary study of ADH2 from *L. fermentati*, it is of great interest to describe the mechanism of ADH2 activation and which type of gene regulations that mostly affect the transcription process. Through this knowledge, the probable use of the ADH2 gene in *L. fermentati* as a promoter in a facilitating ethanol synthesis from highly rich acetic acid medium of lignocellulosic hydrolysates could be significant.

### Funding

This study was funded by the Fundamental Research Grant Scheme (Grant No: 02-03-11-1008FR). Norhayati Yaacob was a recipient of Graduate Research Fellowship (GRF) and MyMaster Scholarship (MyBrain 15, Ministry of Higher Education, Malaysia). The funders had no role in study design, data collection and analysis, decision to publish, or preparation of the manuscript.

### Grant Disclosures

The following grant information was disclosed by the authors:
Fundamental Research Grant Scheme: 02-03- 11-1008FR.
Graduate Research Fellowship (GRF).
MyMaster Scholarship.

### Competing Interests

The authors declare there are no competing interests.

### Author Contributions

- Norhayati Yaacob conceived and designed the experiments, performed the experiments, analyzed the data, wrote the paper, prepared figures and/or tables, reviewed drafts of the paper.
- Mohd Shukuri Mohamad Ali and Nor Aini Abdul Rahman contributed reagents/materials/analysis tools, reviewed drafts of the paper.
- Abu Bakar Salleh reviewed drafts of the paper.

### DNA Deposition

The following information was supplied regarding the deposition of DNA sequences:
KU203771.

### Data Availability

Raw data was included in the article.
## Supplemental Information

Supplemental information for this article can be found online at http://dx.doi.org/10.7717/peerj.1751#supplemental-information.

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
