# Peer review of "Effects of glucose, ethanol and acetic acid on regulation of ADH2 gene from Lachancea fermentati"

_PeerJ, doi:10.7717/peerj.1751_

## Round 0.1 · original submission · Major Revisions

Dear Authors,

The delays in reviewing process is due to the fact that a large number of reviewers (12) have declined to review this manuscript. Never-the-less, based on two reviewers comment the status of the Manuscript is now a "major revision". In addition to these two reviewers, personally I have checked your manuscript and feel that some more in silico and experimental characterization are actually essential. We encourage you to submit your revised version.

·

Basic reporting

In this manuscript Yaacob et al., described that ADH2 gene in L. fermentati can be utilized as a promoter for the regulation under ethanol and organic acid stress. As mentioned in the manuscript that not all yeast alcohol dehydrogenase 2 (ADH2) are repressed by glucose. Although Saccharomyces cerevisiae ADH2 gene is regulated by glucose, Pichia stipitis ADH2 gene is regulated by oxygen instead of glucose, whereas Kluyveromyces marxianus ADH2 is regulated by neither glucose nor ethanol. In this manuscript Yaacob et al., nicely demonstrated that ADH2 gene in L. fermentati is regulated by ethanol and organic acids. All the experimental works in this manuscript is carried out appropriately. In order to improve the quality of the manuscript and the viewer-ship I suggest the followings:
1. The authors should summarize their funding in a schematic diagram.
2. The authors should remove all the references from the manuscript that are not pubmed indexed.
3. In order to provide further proof the authors should clone another gene (eg., GFP, GAPDH etc) under the control of ADH2 promoter and study the activation/repression of that gene in presence of glucose/ethanol/organic acids to confirm the role of ADH2 promoter.

Experimental design

In order to provide further proof the authors should clone another gene (eg., GFP, GAPDH etc) under the control of ADH2 promoter and study the activation/repression of that gene in presence of glucose/ethanol/organic acids to confirm the role of ADH2 promoter.

Validity of the findings

The study is still preliminary

Additional comments

The author should validate the function of ADH2 gene promoter with further experiments and remove all the citations from the reference list that are not pubmed indexed.

·

Basic reporting

Basic reporting is standard

Experimental design

Standard

Validity of the findings

data is robust and properly controlled, acceptable

Additional comments

I found the current study on effect assessments of glucose, ethanol and acetic acid ADH2 from Lachancea fermentati. However, I have some points for improvements:
a) Title can be modified from
"Effects of glucose, ethanol and acetic acid on Lachancea
fermentati ADH2 gene regulation"

to "Effects of glucose, ethanol and acetic acid on regulation of ADH2 gene from Lachancea fermentati "


b) Status in other yeast or fungi will be useful - hence change figures 2 & 5 by adding sequences of other species.

c) A phylogenetic tree of this gene will also improve manuscripts quality.

---

## Round 0.2 · Minor Revisions

Dear Authors,

Your manuscript has been re-reviewed by two independent reviewers and based on their comments the manuscript is now considered to be as in "minor revision" state. I insist that you go through the comments very carefully and address the comments raised by one of the reviewer. I am sure that the comments will improve your manuscript significantly.

With best regards.

·

Basic reporting

The content of this manuscript is substantially improved compared to its previous version. However, the authors should pay attention, carefully edit the manuscript and resubmit. It is necessary to reframe several sentences within the manuscript.

Experimental design

In my opinion there is no further requirement for additional experiments.

Validity of the findings

The findings are valid

Additional comments

The content of this manuscript is substantially improved compared to its previous version. In my opinion there is no further requirement for additional experiments. However, the authors should pay attention, carefully edit the manuscript and resubmit. It is necessary to reframe several sentences within the manuscript. Few examples are:
1. “……..no stranger to the study of alcohol dehydrogenase (ADH) gene…..”
2. “These half-sized yeast chromosomes descended from…………..”
3. This protein would then be compared to the expression of commercially available yeast, Saccharomyces cerevisiae ADH2.
4. “…………….such as high temperature, ethanol inhibition and osmotic pressure…..”
5. “ Hence, the genes that are categorized in the functional categories of
“aerobic respiration” and “mitochondrial function” are targeted for its role in ethanol and organic acid tolerance.
6. Alcohol dehydrogenase 2 (ADH2) is a gene that is negatively regulated by glucose. Due to this, ADH2 can become an important marker in determining the glucose, ethanol and acetic acid stress responses.
7. Line 156: provide space between 20 and μl.
8. “………………not commonly utilized for industrial use.”
9. Denis et al…………write et al in italics.
10. Line 297: Write “Saccharomyces. pombe………..” as “S. pombe………..”

I suggest the followings:
1. Carefully edit the manuscript to remove all the grammatical and topological errors.
2. Shorten the manuscript (keep all the results), but remove describing the standard protocols from the method section.
3. Write the full name of all the organisms for the first time and there after using standard abbreviations. For example write Saccharomyces cerevisiae for the first time and thereafter write as S. cerevisiae. Write hour as hr or h; minutes as mins etc.
I will be happy to re-review the manuscript.

·

Basic reporting

OK

Experimental design

OK

Validity of the findings

OK

Additional comments

No other issues

---

## Round 0.3 · accepted · Accept

However, I insist that you check your manuscript once again for minor mistakes, such as all scientific names, fonts (italics and usages of space) etc. Our PeerJ staffs will also instruct you if the figures have sufficient resolution suitable for printing the on-line files.